# A System in Package Based on a Piezoelectric Micromachined Ultrasonic Transducer Matrix for Ranging Applications

**DOI:** 10.3390/s21082590

**Published:** 2021-04-07

**Authors:** Alexandre Robichaud, Dominic Deslandes, Paul-Vahé Cicek, Frederic Nabki

**Affiliations:** 1Department of Applied Sciences, Université du Québec à Chicoutimi (UQAC), Chicoutimi, QC G7H 2B1, Canada; 2Department of Electrical Engineering, École de Technologie Supérieure (ETS), Montreal, QC H3C 1K3, Canada; Dominic.Deslandes@etsmtl.ca (D.D.); Frederic.Nabki@etsmtl.ca (F.N.); 3Department of Computer Sciences, Université du Québec à Montréal (UQAM), Montreal, QC H2X 3Y7, Canada; cicek.paul-vahe@uqam.ca

**Keywords:** piezoelectric micromachined ultrasonic transducers (PMUT), high voltage pulser, transimpedance amplifier (TIA), regulated cascode (RGC)

## Abstract

This paper proposes a system in package (SiP) for ultrasonic ranging composed of a 4 × 8 matrix of piezoelectric micromachined ultrasonic transducers (PMUT) and an interface integrated circuit (IC). The PMUT matrix is fabricated using the PiezoMUMPS process and the IC is implemented in the AMS 0.35 µm technology. Simulation results for the PMUT are compared to the measurement results, and an equivalent circuit has been derived to allow a better approximation of the load of the PMUT on the IC. The control circuit is composed of a high-voltage pulser to drive the PMUT for transmission and of a transimpedance amplifier to amplify the received echo. The working frequency of the system is 1.5 MHz.

## 1. Introduction

Because conventional ultrasonic transducers are subject to several drawbacks in terms of cost and efficiency [1], micromachined ultrasonic transducers (MUT) have garnered considerable interest in the research community, due to their potential for integration with integrated circuits (IC) and their ability to be mass produced [2,3,4]. There are two types of MUTs: piezoelectric micromachined ultrasonic transducers (PMUT) and capacitive micromachined ultrasonic transducers (CMUT).

For actuation purposes, CMUTs require DC electrostatic biasing, often at a level near membrane pull-in [5]. Furthermore, to achieve full acoustic sensing, multiple devices with different transduction air gap sizes must be realized concurrently in order to achieve strong acoustic signal transmission (large air gap) and sensitive reception (small air gap) [6]. Contrary to CMUTs, no DC biasing is required to operate PMUTs, while they generally also exhibit superior signal-to-noise ratio [6]. PMUTs have been used, among others, to implement devices for distance sensing [7,8], medical imaging [9,10,11] and gesture recognition [12,13].

This work presents the design of a complete ultrasonic imaging system in package (SiP) relying on a PMUT matrix. A 4 × 8 PMUT matrix and its interface CMOS (Complementary Metal Oxide Semiconductor) IC are combined in a single package resulting in high performance, compact size, low cost, and reduced power consumption. Ultimately, the combination of the advances presented in this work could lead to high performance monolithically integrated piezoelectric ultrasonic sensing systems. The transducer matrix is designed with finite element analysis software, followed by model extraction of equivalent lumped electrical elements, which are used as loads in the electronic design of the interface IC. In Section 2, an overview of the overall system is presented, followed by the PMUT design, fabrication and characterization in Section 3. Finally, Section 4 covers the design of the CMOS interface chip. More details about the PMUT design can be found in previous publications of the authors [14,15,16].

## 2. System Overview

The proposed SiP is composed of a PMUT matrix, fabricated using the PiezoMUMPS [17] microfabrication technology, along with its interface CMOS IC for signal driving and sensing, as presented in Figure 1. The PMUTmatrix, with its eight columns of four PMUTs, is able to perform beamforming by driving each element with properly time-delayed electrical pulses. For this proof-of-concept design, the PMUT of a column are electrically connected together, an approach that was selected in order to reduce the number of channels on the CMOS die to 8, hence minimizing chip size and cost. Therefore, the matrix here is only addressable column-wise and beamforming is only possible in one dimension. The reader should note that this constitutes a prototyping design choice rather than a technical limitation. Finally, in this work the functionality of the SiP is verified based on a single channel.

The CMOS circuit, designed in high voltage 0.35 μm technology from AMS (Austria Mikro Systeme), is composed of pulsers and transimpedance amplifiers (TIA). Each column of the PMUT matrix requires a pulser and a TIA, for a total of eight cells.

The role of a pulser is to drive a PMUT with an electrical excitation of sufficient amplitude to generate a strong acoustic pulse. This system provides an amplitude up to 50 $V_{\mathrm{pp}}$, the maximum allowed by the CMOS technology. As for the TIA, its purpose is to sense and amplify the returning acoustic echo to allow for accurate sampling. For each matrix column, a distinct pulser-TIA interface cell is connected to all of the PMUTs of that set. To avoid that the high-voltage output of the pulser causes any damage to the TIA in the proposed system, the TIA could be decoupled from the circuit by a switch during acoustic transmission. For the purposes of this prototype, distinct columns of PMUTs were used to transmit and receive, avoiding any potential damage to the TIA.

For this prototype, an Agilent 332320A arbitrary waveform generator is used to activate the pulsers, modulate the desired beamforming, and sample the echo back. In future work, this functionality is to be accomplished by the CMOS circuit.

### Fabrication Process

As mentioned before, the PMUTs were fabricated using the PiezoMUMPS process [17]. It is a low-cost process involving five masks. The process uses an SOI wafer on which a layer of aluminum nitride (AlN) and aluminum (Al) are deposited and patterned. The AlN layer is 500 nm thick and is used as piezoelectric material. According to the PiezoMUMPS design handbook, the AlN used has a piezoelectric strain coefficient, d33, in the order of 3.4–6.5 (pC/N). The aluminum layer is 1 μm thick and serves as the top electrode. The membrane (substrate) is 400 μm thick and serves as the bottom electrode. In order to release the membrane, the DRIE process is used to etch from the back of the SOI wafer. The pad oxide is 200 nm thick and the total thickness of the PMUT is 412.7 μm. The result is a membrane having a diameter of 200 μm suspended by four anchoring arms. The Al and AlN layer have a smaller diameter in order to respect the design rules of the technology. Figure 2 shows a schematic of the device.

A matrix of 4 × 8 PMUTs was fabricated. Figure 3a shows the fabricated PMUT matrix and Figure 3b shows a detailed view of a single PMUT.

## 3. Design, Fabrication and Characterization of the PMUT Matrix

### 3.1. PMUT Equivalent Circuit

A PMUT can be represented by the Butterworth-Van Dyke equivalent circuit presented in Figure 4. The capacitance $C_{0}$ is the electrical capacitance of the PMUT and can be estimated using this equation:(1)$$C_{0}=\epsilon \frac{A}{d},$$
where *A* is the area of the PMUT, ϵ is the dielectric constant of the AlN layer and *d* is the thickness of the AlN layer. The device was fabricated using the PiezoMUMPS technology which uses AlN as piezoelectric layer. The parameters $L_{m}$, $C_{m}$ and $R_{m}$ represent the mechanical properties of the structure. They correspond to the mass, stiffness and damping, respectively, and are expressed in terms of electrical units (H, F and Ω).

Finally, parameters $L_{r}$ and $R_{r}$ correspond to the real and imaginary parts of the acoustic impedance, respectively. Techniques to calculate the values of the lumped parameters can be found in [18,19,20].

### 3.2. Simulations

To undertake the design of the PMUT, COMSOL Multiphysics was used in combination with the physical parameters of the PiezoMUMPS process as specified in the design handbook of the technology. The modeled PMUT is composed of a silicon membrane of 100 μm radius and 10 μm thickness on top of which a 500 nm AlN layer and a 1 μm aluminum layer are deposited. These layers act as the piezoelectric layer and the top electrode, respectively. The membrane is suspended by four supporting arms. As compared to a fully anchored membrane, the stiffness of the structure is reduced resulting in an increase of the displacement of the membrane and of the acoustic power.

Finally, the membrane is incorporated in a sphere of air to take into consideration air damping and obtain more accurate results. Although this system was designed for operation in air, it is expected to be suitable for operation within liquid media. As observed in some preliminary tests, this can lead to a longer transmission range. However, if the medium is conductive, an electrically insulating layer such as Parylene would need to be added above the device. The radius of the sphere is 500 μm. The air sphere was split into two halves, each positioned on the top and bottom of the PMUT, effectively separating the two half spheres by a distance equal to the thickness of the PMUT. A sound hard boundary condition was set on the flat inner side of the half sphere and a far-field calculation was added on the surface of the sphere as shown in Figure 5a. The mesh was built using free tetrahedrals, with a fine mesh used for the PMUT and a coarser mesh for the sphere. A terminal with a voltage amplitude of 1 V was applied to the PMUT. Finally, a fixed constraint condition was set at the end of each anchoring arm of the membrane. Figure 5b shows acoustic pressure simulation results.

To identify the device resonant frequency, eigenmode simulations were undertaken. The frequency of the first mode was found to be at around 1.5 MHz. Simulations and measurements of the shape of this mode have been undertaken in previous work, demonstrating an effective coupling to air despite the supporting arms [14,15,16].

### 3.3. Characterization

A PMUT was characterized using an E5061B Keysight PNA Network Analyzer, and an EP6 Cascade Probe Station, in combination with GSG (ground-signal-ground) probes. Conductance was measured by sweeping the frequency from 1.3 to 1.7 MHz, as presented in Figure 6.

As a first step, to estimate the values of the lumped parameters of the equivalent circuit described in Section 3.1, the equations in [18,19,20] were used. Based on the obtained circuit the conductance as a function of frequency was calculated. Finally, curve fitting was made by adjusting the parameters to obtain a match between calculation and measurement. This allows extracting an accurate load model to perform the design of the CMOS chip. The values of the lumped elements obtained for $L_{m}$, $C_{m}$, $R_{m}$, $R_{r}$, $L_{r}$, $C_{0}$ and N are 902 pH, 12.6 μF, 25.6 μΩ, 14.3 μΩ, 0.36 pH, 5.72 pF and 3.7e-5 respectively.

## 4. Design of the CMOS Chip

The CMOS interface chip is composed of several identical high-voltage pulsers and TIAs, realized in high-voltage AMS 0.35 μm CMOS technology. The high-voltage pulser is driven by a square wave with an amplitude of 3 V at its input, providing at its output a square wave of up to 50 $V_{\mathrm{pp}}$, with the same frequency and duty cycle. The TIA amplifies the electrical signal produced by a PMUT in response to an acoustic echo, with a gain of 87 dBΩ and a 3-dB bandwidth of 12 MHz.

### Design of the High-Voltage Pulser

To amplify the echo signal, a regulated cascode (RGC) TIA was chosen. It provides a small input impedance and a high bandwidth. Figure 7a shows a schematic of the circuit. The input signal is connected to the source of a common gate (CG) amplifier. The input is also connected to a common source (CS) amplifier. The output of the CS is connected back to the CG forming a feedback which decreases the input impedance by a factor proportional to the gain of the CS. The input impedance is given by:(2)$$Z_{in}\approx \frac{1}{g_{m,1}\cdot \left(1+g_{m,2}\cdot 1/g_{m,3}\right)},$$
where $1/g_{m,2}$ and $1/g_{m,3}$ are the transconductance of transistor M2 and M3. and the gain is given by:(3)$$Z_{T}\approx 1/g_{m,4},$$
where $1/g_{m,4}$ is the transconductance of transistor M4. The pad to the input of the RGC is to be placed as close as possible to its corresponding PMUT to eliminate the parasitic capacitance and thus minimize the signal loss due to charge sharing, and thus increase the overall SNR. The electrical connection is made using a gold wire-bond. Figure 8 shows the schematic of the wire-bond connections between the CMOS chip and the PMUT array.

We have not performed a detailed analysis of the minimum detectable signal. However, it is expected that a combination of the input referred electronic noise of the TIA in combination with the mechanical noise of the PMUT itself would consist in the ultimate limit, which would be degraded in the presence or significant ambient acoustic noise. We expect that the electronic noise of the TIA would dominate, provided that the ambient acoustic noise is sufficiently low.

The pulser is composed of a level shifter followed by a high-voltage driver. Figure 7b shows the circuit. The 3.3-V input signal is applied at the gate of M1. When $V_{in}$ is 0 V, the voltage at the drain of M4 (source of M3) is equal to $V_{DD}$ (up to 50 V). When $V_{in}$ is equal to 3.3 V, M1 becomes a short-circuit. In this case, diode-connected transistors M2 and M3 behave as a voltage divider with its output taken at the drain of M3 and applied at the gate of M4, thus turning it on. Sizing of the transistors M2-M3 was selected so that the voltage at the drain of M3 varies between approximately 50 V and 45 V allowing to fully turn the high-voltage transistor M4 on and off respectively. $V_{in}$ is also applied to the input of the inverter implemented using M5 and M6, applying the inverse of $V_{in}$ to the gate of M7. Hence, when $V_{in}$ is high, M7 is turned off and M4 is turned on such that $V_{out}$ becomes a high-voltage replica of $V_{in}$.

## 5. Measurement Results

The RGC TIA sensing amplifier was tested using an oscilloscope, with the resulting transfer function presented in Figure 9a. The measured gain is about 64 dBΩ and the measured bandwidth is about 1.5 MHz.

The high-voltage pulser was tested using an oscilloscope. The results are shown in Figure 9b. One can see that the minimum output voltage corresponds to the $V_{tn}$ of M5. Although the specified working frequency of 1.5 MHz is achieved, the signal suffers slight distortion, which can be attributed to the slew rate of the TIA output. Similarly, the bandwidth is reduced with respect to simulation expectations. These discrepancies can be attributed to the underestimation in simulations of the capacitive parasitic effects due to the packaging interconnection with the PMUT device. It is important to mention that the final target system is expected to integrate the PMUT directly on the CMOS chip which will then be subject to minimal parasitic effects.

An analysis of the minimum detectable signal was not performed. However, it is expected that a combination of the input referred electronic noise of the TIA in combination with the mechanical noise of the PMUT itself would consist in the ultimate limit, which would be degraded in the presence or significant ambient acoustic noise. We expect that the electronic noise of the TIA would dominate, provided that the ambient acoustic noise is sufficiently low.

### Design of the Transimpedance Amplifier

The fully realized system was used to perform ranging measurements. The PCB was mounted on a translation stage from the company THORLABS. A square wave signal was used to drive the high-voltage pulser. The signal is 4 period-long and has an amplitude of 3.3 V and a frequency of 1.5 MHz. A supply voltage of 20 V was used for the pulser. The pulser was wire-bonded to a first column of PMUTs in order to generate acoustic waves. The reflected signal was then received by an adjacent column of PMUT which was bonded to the RGC TIA. The stage was adjusted to vary the transmission distance d between the chip and the reflector (a flat sheet of copper). Before undertaking the measurements, the PMUT were calibrated in order to minimize the influence of the cross-talk. As illustrated in Figure 10, this was achieved by measuring and characterizing the steady-state cross-talk between the emitting and receiving PMUT by sending a test pulse prior to transmission (red), then systematically subtracting that baseline from the ranging measurements (blue), in order to accurately represent the intended acoustic signal (orange).

Figure 11 shows the measurement setup. The output of the RGC was measured using an oscilloscope. Figure 12 shows the measurements results for distances *d* of 2 mm, 4 mm, 6 mm, 8 mm, 10 mm and 12 mm. This shows that the system can effectively be used to undertake ranging measurement with excellent accuracy. The echo is a very weak signal. Therefore, the TIA was shown to have sufficient gain to amplify the signal in order for it to be perceptible at the output.

## 6. Conclusions

This paper presented a SiP composed of a matrix of PMUTs and their interface CMOS circuit. An equivalent circuit for a PMUT was established and then used as the load for the simulation of the CMOS circuit. The high-voltage pulser was designed to drive the PMUT and simulations showed that it was capable of doing so with a 20 $V_{\mathrm{pp}}$ square wave at 1.5 MHz. A TIA was designed, fabricated and used to amplify the echo received. The fully realized SiP was successfully used to undertake ranging measurements at up to 12 mm. The results have shown that the SiP can be used to undertake this task and represents a compact implementation that can be the basis for an imaging system in ulterior prototypes building on this work. Deeper investigation of the ultrasonic behavior of the PMUT matrix, with and without beamforming control, would be of interest to further the understanding of ultrasonic ranging methods, and is planned as future work.

## Figures and Tables

**Figure 1 sensors-21-02590-f001:**
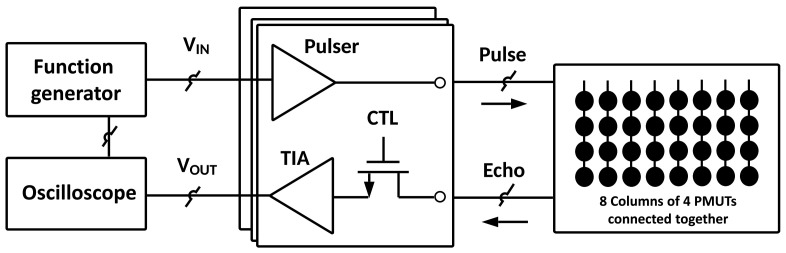
Block diagram of the proposed PMUT SiP.

**Figure 2 sensors-21-02590-f002:**
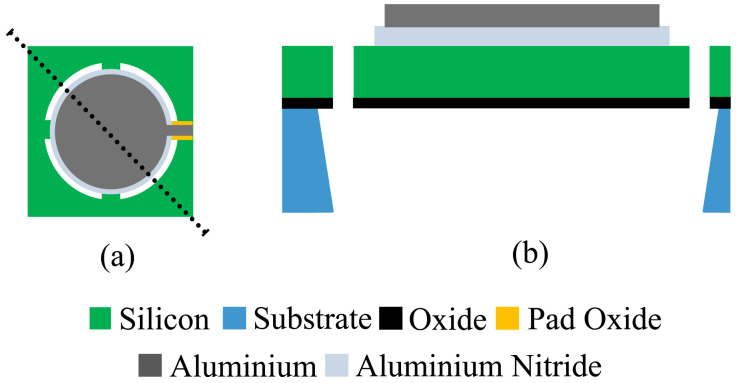
(**a**) Top view and (**b**) Cross Section of PMUT fabricated using the PiezoMUMPS technology.

**Figure 3 sensors-21-02590-f003:**
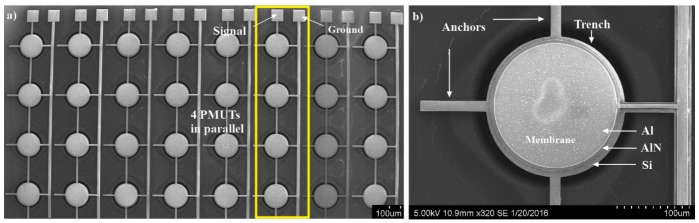
Micrographe of (**a**) the fabricated PMUT matrix and (**b**) of a single PMUT.

**Figure 4 sensors-21-02590-f004:**
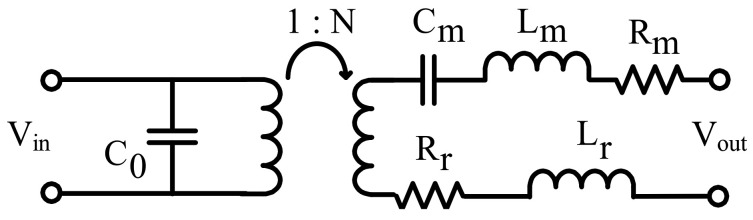
Equivalent circuit of a PMUT.

**Figure 5 sensors-21-02590-f005:**
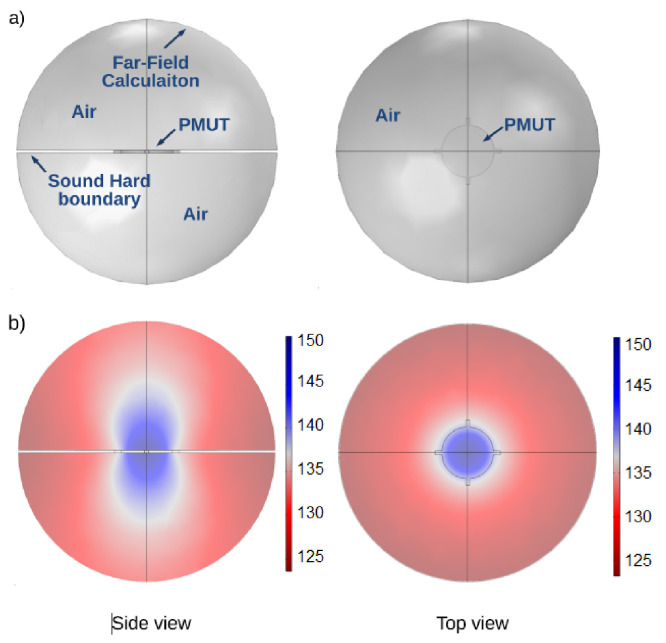
COMSOL simulation. (**a**) model used for the simulations and (**b**) simulation results of acoustic pressure level in dBV.

**Figure 6 sensors-21-02590-f006:**
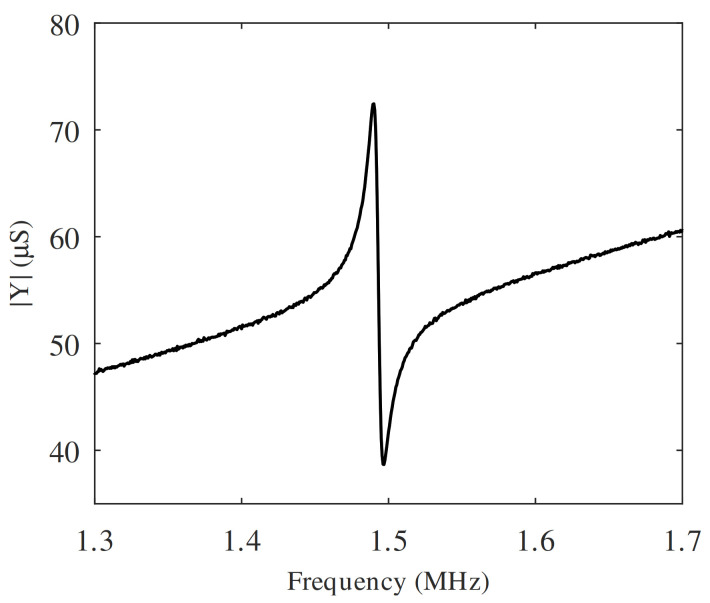
Measurement of the admittance of a PMUT as a function of frequency.

**Figure 7 sensors-21-02590-f007:**
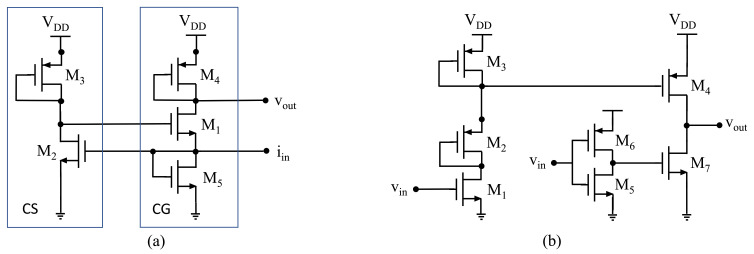
Schematic of (**a**) the TIA amplifier and (**b**) the high-voltage pulser.

**Figure 8 sensors-21-02590-f008:**
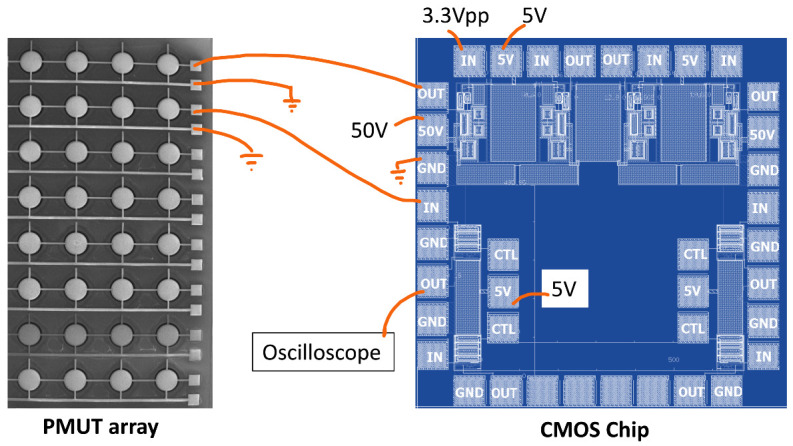
Schematic of the wire-bond connections between the CMOS chip and the PMUT array.

**Figure 9 sensors-21-02590-f009:**
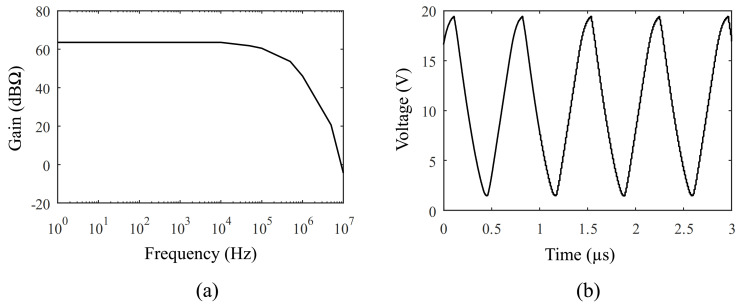
Measurement results: (**a**) Frequency domain measurements of the RGC TIA, (**b**) Time domain measurements of the high-voltage pulser.

**Figure 10 sensors-21-02590-f010:**
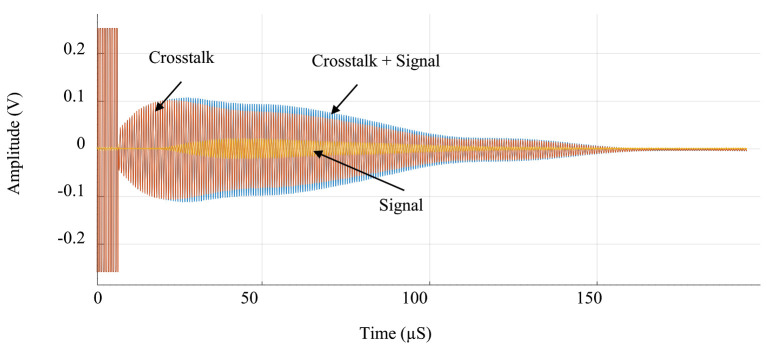
Measurement of the crosstalk.

**Figure 11 sensors-21-02590-f011:**
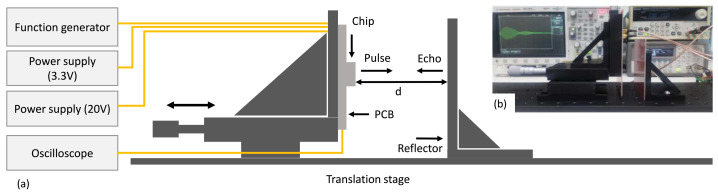
(**a**) Schematic and (**b**) photograph of the measurement setup.

**Figure 12 sensors-21-02590-f012:**
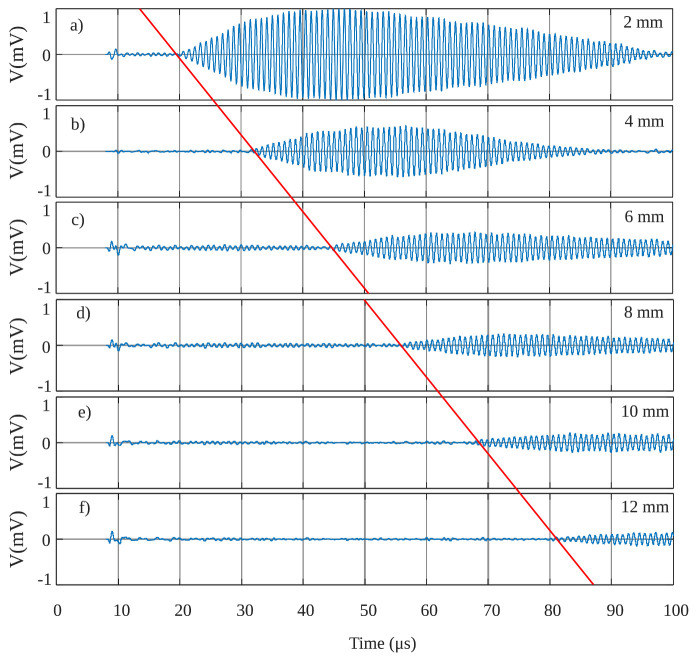
Ranging measurement results in the air for distances of (**a**) 2 mm, (**b**) 4 mm, (**c**) 6 mm, (**d**) 8 mm, (**e**) 10 mm, and (**f**) 12 mm.

## Data Availability

Not applicable.

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
