# Peer review of "A System in Package Based on a Piezoelectric Micromachined Ultrasonic Transducer Matrix for Ranging Applications"

_sensors, 2021, doi:10.3390/s21082590_

Round 1

Reviewer 1 Report

This work, titled “A System in Package Based on a Piezoelectric Micromachined Ultrasonic Transducer Matrix for Ranging Applications”, reports and proposes a system in package (SiP) for ultrasonic ranging composed of a 4 × 8 matrix of piezoelectric micromachined ultrasonic transducers (PMUT) based on Aluminum Nitride piezoelectric material and an interface integrated circuit (IC). A high-voltage pulser was designed and included in the driving system to drive the PMUT whose operating frequency is designed to be at 1.5 MHz. The fully-realized SiP was successfully used to launch ranging measurements at up to 12 mm and it demonstrates an implementation that can be the basis for a compact imaging system.

The reviewer found this work interesting and original. However, revisions are needed before being further considered for publication in Sensors. Please, address the following comments:

  1. The proposed SiP is composed of a 4x8 matrix of pMUT based on AlN piezoelectric thin film and using aluminium as top electrode. From fabrication schematic, it seems that no bottom electrodes were used as ground. Have the membranes of the array been embedded in two top and bottom electrodes? If so, which is the thickness of the ground electrode?

  1. Aluminum Nitride has been used as piezoelectric material. Has it been characterized in terms of piezoelectric features? Which are the piezoelectric coefficients and the electromechanical coupling coefficient of the material?

  1. The working frequency of the pMUT has been fixed at 1.5 MHz. Does the pMUT exploit the resonance of the first resonant mode? If so, how do the anchoring arms affect or change the piston-like movement of the pMUT?

  1. At line 142 of the paragraph “ Measurement Results”, the authors state the TIA introduces a slight distortion due to the slew rate. Does this distortion affects the performances of the system, also in terms of working frequency of the pMUT?

  1. At line 155 of the subparagraph “Design of the Trasimpedance Amplifier”, the authors discuss about pMUT calibration to minimize the influence of cross-talk. How has this calibration been performed and how the cross-talk has been estimated and taken into account?

  1. The SiP device consists of a 4x8 matrix of pMUT whose final goal is to perform the beamforming. How the beamforming improve the performances of the SiP? A comparison of the behaviour of the matrix with and without beamforming control could be useful.

  1. Have the directivity and radiation pattern of the SiP been analysed at different measuring distances?

Author Response

We are very grateful to the editor for arranging a careful review of this revised work, and to the reviewers for the time and effort spent in evaluating this manuscript. The constructive comments of the reviewers are highly appreciated, and their detailed suggestions have provided us with an opportunity to improve the manuscript further. We have considered these comments carefully and the manuscript has been revised to address them.

The proposed SiP is composed of a 4x8 matrix of pMUT based on AlN piezoelectric thin film and using aluminum as top electrode. From fabrication schematic, it seems that no bottom electrodes were used as ground. Have the membranes of the array been embedded in two top and bottom electrodes? If so, which is the thickness of the ground electrode?

Thank you for your comments.

The following was added to section 3.2 to clarify:

"The aluminum layer is 1 μm thick and serves as the top electrode. The membrane is 400 μm thick and serves as the bottom electrode."

Aluminum Nitride has been used as piezoelectric material. Has it been characterized in terms of piezoelectric features? Which are the piezoelectric coefficients and the electromechanical coupling coefficient of the material?

The piezoelectric material has not been independently characterized in terms of piezoelectric properties. However, the PiezoMUMPS design handbook provides some useful information which was added to section 3.2:

 "According to the PiezoMUMPS design handbook, the AlN used in the technology has a piezoelectric strain coefficient, d33, in the order of 3.4-6.5 pC/N."

The working frequency of the pMUT has been fixed at 1.5 MHz. Does the pMUT exploit the resonance of the first resonant mode? If so, how do the anchoring arms affect or change the piston-like movement of the pMUT?

The following was added to section 3.3 :

"To identify the device resonant frequency, eigenmode simulations were undertaken. The frequency of the first mode was found to be at around 1.5 MHz. Simulations and measurements of the shape of this mode have been undertaken in previous work, demonstrating an effective coupling to air despite the supporting arms [15–17]."

At line 142 of the paragraph “ Measurement Results”, the authors state the TIA introduces a slight distortion due to the slew rate. Does this distortion affects the performances of the system, also in terms of working frequency of the pMUT?

The following was added to section 5 :

"Although the specified working frequency of 1.5 MHz is achieved, the signal suffers slight distortion, which can be attributed to the slew rate of the TIA output. Similarly, the bandwidth is reduced with respect to simulation expectations. These discrepancies can be attributed to the underestimation in simulation of the capacitive parasitic effects due to the packaging interconnection with the PMUT device. It is important to mention that the final target system is expected to integrate the PMUT directly on the CMOS chip which will then be subject to minimal parasitic effects."

At line 155 of the subparagraph “Design of the Transimpedance Amplifier”, the authors discuss about pMUT calibration to minimize the influence of cross-talk. How has this calibration been performed and how the cross-talk has been estimated and taken into account?

 The following was added to section 5.1 :

" this was achieved by measuring and characterizing the steady-state cross-talk between the emitting and receiving PMUT by sending a test pulse prior to transmission (red), then systematically subtracting that baseline from the ranging measurements (blue), in order to accurately represent the intended acoustic signal (orange).” 

The SiP device consists of a 4x8 matrix of pMUT whose final goal is to perform the beamforming. How the beamforming improve the performances of the SiP? A comparison of the behavior of the matrix with and without beamforming control could be useful.

Have the directivity and radiation pattern of the SiP been analyzed at different measuring distances?

We agree that deeper investigation of the behavior of the matrix with and without beamforming would be of great interest to further explore the possibilities afforded by this approach. However, we believe that such an analysis goes beyond the immediate scope of reporting on the experiment carried-out in this work, which presents a path towards implementing a practical system-in-package. As a next step, this system will provide the opportunity to investigate beamforming in the field.

To address the last two points, the following was added to the conclusion to outline the potential for future work:

"Deeper investigation of the ultrasonic behavior of the PMUT matrix, with and without beamforming control, would be of interest to further the understanding of ultrasonic ranging methods, and is planned as future work."

Reviewer 2 Report

Dear author,

I found your work interesting and clearly explained. I have a few comments and curiosities:

  • You could enrich the design session of the device with a few details on the simulation conditions. Like what voltage was applied in the simulation to obtain the acoustic pressure. Moreover, it would be nice to have a comparison between the experimental results and the expected acoustic signal.
  • You designed a pulser to provide 50Vpp. Although already at 20Vpp the signal is distorted by the slew rate. What is the theoretical slew rate of your pulser? Could you comment on this point?
  • Also the TIA shows performance slightly off design. Your bandwidth is reduced by a factor 10 and also your gain is lower. Do you have an explanation for the loss of performance?
  • You mentioned a switch to protect the TIA from the pulser, but you actually used two different channels for sensing and actuation. Any comment on this point?
  • Do you have an idea on the minimum detectable signal i.e. range detection limit? What is the ultimate limit for the measurements? Is it the equivalent acoustic noise of the TIA or background acoustic noise?

Author Response

We are very grateful to the editor for arranging a careful review of this revised work, and to the reviewers for the time and effort spent in evaluating this manuscript. The constructive comments of the reviewers are highly appreciated, and their detailed suggestions have provided us with an opportunity to improve the manuscript further. We have considered these comments carefully and the manuscript has been revised to address them.

You could enrich the design session of the device with a few details on the simulation conditions. Like what voltage was applied in the simulation to obtain the acoustic pressure. Moreover, it would be nice to have a comparison between the experimental results and the expected acoustic signal.

Thank you for your comments.

The following was added to section 3.2:

"The radius of the sphere is 500 µm. The air sphere was split into two halves, each positioned on the top and bottom of the PMUT, effectively separating the two half spheres by a distance equal to the thickness of the PMUT. A sound hard boundary condition was set on the flat inner side of the half sphere and a far-field calculation was added on the surface of the sphere as shown in figure 5 a). The mesh was built using free tetrahedrals, with a fine mesh used for the PMUT and a coarser mesh for the sphere. A terminal with a voltage amplitude of 1 V was applied to the PMUT. Finally, a fixed constraint condition was set at the end of each anchoring arm of the membrane."

You designed a pulser to provide 50Vpp. Although already at 20Vpp the signal is distorted by the slew rate. What is the theoretical slew rate of your pulser? Could you comment on this point?

Also the TIA shows performance slightly off design. Your bandwidth is reduced by a factor 10 and also your gain is lower. Do you have an explanation for the loss of performance?

To address the last two points, the following was added to section 5.

"Although the specified working frequency of 1.5 MHz is achieved, the signal suffers slight distortion, which can be attributed to the slew rate of the TIA output. Similarly, the bandwidth is reduced with respect to simulation expectations. These discrepancies can be attributed to the underestimation in simulation of the capacitive parasitic effects due to the packaging interconnection with the PMUT device. It is important to mention that the final target system is expected to integrate the PMUT directly on the CMOS chip which will then be subject to minimal parasitic effects."

You mentioned a switch to protect the TIA from the pulser, but you actually used two different channels for sensing and actuation. Any comment on this point?

The following was modified in section 2 :

"To avoid that the high-voltage output of the pulser causes any damage to the TIA in the proposed system, the TIA could be decoupled from the circuit by a switch during acoustic transmission. For the purposes of this prototype, distinct columns of PMUTs were used to transmit and receive, avoiding any potential damage to the TIA."

Do you have an idea on the minimum detectable signal i.e. range detection limit? What is the ultimate limit for the measurements? Is it the equivalent acoustic noise of the TIA or background acoustic noise?

The following was added to section 5 : 

"An analysis of the minimum detectable signal was not performed. However, it is expected that a combination of the input referred electronic noise of the TIA in combination with the mechanical noise of the PMUT itself would consist in the ultimate limit, which would be degraded in the presence or significant ambient acoustic noise. We expect that the electronic noise of the TIA would dominate, provided that the ambient acoustic noise is sufficiently low."

Reviewer 3 Report

The paper is devoted to development of a system in package comprising set of piezoelectric micromachined ultrasonic transducers (PMUT) and an interface integrated circuit. This paper is in scope of Sensors and could be considered for publication after major revision.

  1. It is necessary to expand significantly the Introduction. It is not understandable what new in this paper. Ref [5] is not devoted to CMUT as authors pointed. This paper is devoted to PMUT and no any discussion about ВС biasing, etc. in this paper. There is no information about the air gap size influence in Ref [6]. Check, please.
  2. My suggestion is to reorganize paper. It will be good to show 4x8 matrix of PMUTs as authors say about in first time. Now it is not understandable what means 8 columns at line 40.
  3. It will be good to exchange items 3.1 and 3.2. Now authors talk about AlN in item 3.1 before explanation where this layer is.
  4. At line 78 is Aln means AlN?
  5. It will be good to add more information about Comsol simulation. What mesh has been used? What boundary conditions have been used? Etc.
  6. At line 111 what is Vpp?
  7. In item 4.1 it is necessary to start description from Fig7(a) and then Fig7(b). The same for Figure 8.
  8. At line 118 what is VDD? It will be good to show it in Figure 7. As well it will be good show in Figure CG and CS.
  9. At line 151. It will be good show in Figure how pulser was bonded to a first column of PMUT.
  10. At lines 157-158 it will be good to introduce the variable l for distance and show in Figure where is it.
  11. In Eq. 2 introduce gm,1-4, Z. Usually Z means impedance.

It will be good to explain the possible application of developed system more detailed. 

Author Response

We are very grateful to the editor for arranging a careful review of this revised work, and to the reviewers for the time and effort spent in evaluating this manuscript. The constructive comments of the reviewers are highly appreciated, and their detailed suggestions have provided us with an opportunity to improve the manuscript further. We have considered these comments carefully and the manuscript has been revised to address them.

It is necessary to expand significantly the Introduction. It is not understandable what new in this paper. Ref [5] is not devoted to CMUT as authors pointed. This paper is devoted to PMUT and no any discussion about ВС biasing, etc. in this paper. There is no information about the air gap size influence in Ref [6]. Check, please.

Thank you for your comments.

The following has been added to the introduction:

“A 4 X 8 PMUT matrix and its interface CMOS (Complementary Metal Oxide Semiconductor) IC are combined in a single package resulting in high performance, compact size, low cost, and reduced power consumption. Ultimately, the combination of the advances presented in this work could lead to high performance monolithically integrated piezoelectric ultrasonic sensing systems.”

The references have been corrected.

My suggestion is to reorganize paper. It will be good to show 4x8 matrix of PMUTs as authors say about in first time. Now it is not understandable what means 8 columns at line 40.

Figure 1 was modified to illustrate the matrix structure at the start of the paper.

It will be good to exchange items 3.1 and 3.2. Now authors talk about AlN in item 3.1 before explanation where this layer is.

The order of Section 3.1 and Section 3.2 was inverted.

At line 78 is Aln means AlN?

This was corrected.

It will be good to add more information about Comsol simulation. What mesh has been used? What boundary conditions have been used?

The following was added to section 3.2:

"The radius of the sphere is 500 µm. The air sphere was split into two halves, each positioned on the top and bottom of the PMUT, effectively separating the two half spheres by a distance equal to the thickness of the PMUT. A sound hard boundary condition was set on the flat inner side of the half sphere and a far-field calculation was added on the surface of the sphere as shown in figure 5 a). The mesh was built using free tetrahedrals, with a fine mesh used for the PMUT and a coarser mesh for the sphere. A terminal with a voltage amplitude of 1 V was applied to the PMUT. Finally, a fixed constraint condition was set at the end of each anchoring arm of the membrane."

 At line 111 what is Vpp?

The sentence was modified to: “The signal is 4 period-long and has an amplitude of 3.3 V.”

In item 4.1 it is necessary to start description from Fig7(a) and then Fig7(b). The same for Figure 8.

The description of Fig7(a) was moved before the description of Fig7(b)

At line 118 what is VDD? It will be good to show it in Figure 7. As well it will be good show in Figure CG and CS.

The value of VDD has been specified:

“When Vin is 0 V, the voltage at the drain of M4 (source of M3) is equal to VDD (up to 50 V).”

Additionally, VDD, CG and CS have been added to figure 7.

At line 151. It will be good show in Figure how pulser was bonded to a first column of PMUT.

A figure was added to section 4.1 to show the connections between the CMOS IC and the PMUT array.

At lines 157-158 it will be good to introduce the variable lfor distance and show in Figure where is it.

A variable was introduced for the distance and was also added to figure 11.

The stage was adjusted to vary the transmission distance d between the chip and the reflector (the reflector is a copper sheet).

In Eq. 2 introduce gm,1-4, Z. Usually Z means impedance.

The variable gm was clearly introduced such that gm,2 and gm,3 and gm,4  are the transconductances of transistors M2, M3 and M4, respectively.

Reviewer 4 Report

  1. Line 64: Include a short statement explaining why AlN was chosen over other materials such as PZT or PVDF. It can be a short statement that justifies the choice of AlN.
  2. Section 3.2 Fabrication Process: Include the thicknesses of the silicon, oxide, and substrate layers in order to know the total thickness of the PMUTS.
  3. Include a scale line in Figure 4(a).
  4. Section 3.3 Simulations: Include a short statement of how this design would change should the PMUTs need to be immersed in liquid. Would you cover the PMUTs with a protective layer (i.e. parylene)? What would the effects be on the PMUTs performance?
  5. Figure 5(b): Make bigger the numbers of the scale bars; they are hard to see.
  6. Line 97: Change ‘A PMUT were characterized’ to ‘A PMUT was characterized’.
  7. Line 103: Change ‘was calculate’ to ‘was calculated’.
  8. Line 135: Change ‘to eliminated’ to ‘to eliminate’.
  9. Figure 8(c) is missing. It Is the same as Figure 6. Double check this issue.
  10. Line 154: Specify the material of the reflector.
  11. Line 155: Was cross-talk measured or at least modeled? Include some cross-talk measurements or simulated results.

Author Response

We are very grateful to the editor for arranging a careful review of this revised work, and to the reviewers for the time and effort spent in evaluating this manuscript. The constructive comments of the reviewers are highly appreciated, and their detailed suggestions have provided us with an opportunity to improve the manuscript further. We have considered these comments carefully and the manuscript has been revised to address them.

Line 64: Include a short statement explaining why AlN was chosen over other materials such as PZT or PVDF. It can be a short statement that justifies the choice of AlN.

Thank you for your comments. The following was added to section 3.1 :

"The device was fabricated using the PiezoMUMPS technology which uses AlN as the piezoelectric layer."

Section 3.2 Fabrication Process: Include the thicknesses of the silicon, oxide, and substrate layers in order to know the total thickness of the PMUTS.

The following was added to section 3.2

"The aluminum layer is 1μm thick and serves as the top electrode. The membrane (substrate) is 400 μm thick and serves as the bottom electrode. In order to release the membrane, the DRIE process is used to etch from the back of the SOI wafer. The pad oxide is 200 nm thick and the total thickness of the PMUT is 412.7 μm."

Include a scale line in Figure 4(a).

A scale has been included.

Section 3.3 Simulations: Include a short statement of how this design would change should the PMUTs need to be immersed in liquid. Would you cover the PMUTs with a protective layer (i.e. parylene)? What would the effects be on the PMUTs performance?

The following statement was added to Section 3.2:

“Finally, the membrane is incorporated in a sphere of air to take into consideration air damping and obtain more accurate results. Although this system was designed for operation in air, it is expected to be suitable for operation within liquid media. As observed in some preliminary tests, this can lead to a longer transmission range. However, if the medium is conductive, an electrically insulating layer such as Parylene would need to be added above the device.”

Figure 5(b): Make bigger the numbers of the scale bars; they are hard to see.

The size of the numbers was increased.

Line 97: Change ‘A PMUT were characterized’ to ‘A PMUT was characterized’.

This has been corrected.

Line 103: Change ‘was calculate’ to ‘was calculated’.

This has been corrected.

Line 135: Change ‘to eliminated’ to ‘to eliminate’.

This has been corrected.

Figure 8(c) is missing. It Is the same as Figure 6. Double check this issue.

The caption for figure 8 has been adjusted.

Line 154: Specify the material of the reflector.

 The following has been added:

"The stage was adjusted to vary the transmission distance d between the chip and the reflector (a flat sheet of copper).

 Line 155: Was cross-talk measured or at least modeled? Include some cross-talk measurements or simulated results.

A figure showing cross-talk measurement was added to the paper. Also the following was added to section 5.1 :

"As illustrated in figure 10, this was achieved by measuring and characterizing the steady-state cross-talk between the emitting and receiving PMUT by sending a test pulse prior to transmission (red), then systematically subtracting that baseline from the ranging measurements (blue), in order to accurately represent the intended acoustic signal (orange)."

Round 2

Reviewer 1 Report

The authors well addressed all the questions. The reviewer suggests to accept the manuscript in the present form for publication in Sensors

Author Response

Thank you, we will send the final version of the manuscript.

Reviewer 3 Report

After corrections paper could be published.

Author Response

Thanks you, we will send the final version of the manuscript.